# Kernel-Based Function Approximation for Average Reward Reinforcement Learning: An Optimist No-Regret Algorithm

**Sattar Vakili**
MediaTek Research
sattar.vakili@mtkresearch.com

**Julia Olkhovskaya**
TU Delft
julia.olkhovskaya@gmail.com

## Abstract

Reinforcement learning utilizing kernel ridge regression to predict the expected value function represents a powerful method with great representational capacity. This setting is a highly versatile framework amenable to analytical results. We consider kernel-based function approximation for RL in the infinite horizon average reward setting, also referred to as the undiscounted setting. We propose an *optimistic* algorithm, similar to acquisition function based algorithms in the special case of bandits. We establish novel *no-regret* performance guarantees for our algorithm, under kernel-based modelling assumptions. Additionally, we derive a novel confidence interval for the kernel-based prediction of the expected value function, applicable across various RL problems.

## 1 Introduction

Reinforcement learning (RL) has demonstrated substantial practical success across a variety of application domains, including gaming (Silver et al., 2016; Lee et al., 2018; Vinyals et al., 2019), autonomous driving (Kahn et al., 2017), microchip design (Mirhoseini et al., 2021), robot control (Kalashnikov et al., 2018), and algorithmic search (Fawzi et al., 2022). This empirical success has prompted deeper investigations into the analytical understanding of RL, especially in complex environments. Over the past decade, significant advances have been made in establishing theoretically grounded algorithms for various settings. In this work, we focus on the infinite horizon average reward setting, also known as the undiscounted setting (Wei et al., 2020, 2021). This setting is particularly well-suited for applications that involve continuing operations not divided into episodes such as load balancing and stock market operations. In contrast to the episodic setting (Jin et al., 2020) and the discounted setting (Zhou et al., 2021), theoretical understanding of RL algorithms is relatively limited for the undiscounted setting. We develop a computationally efficient algorithm and establish its theoretical performance guarantees in the undiscounted case.

There is a natural progression in the complexity of RL models corresponding to the structural complexity of the Markov Decision Process (MDP). This progression ranges from tabular models to linear, kernel-based, and deep learning-based models. The kernel-based structure is an extension of linear structure to an infinite-dimensional linear model in the feature space of a positive definite kernel, resulting in a highly versatile model with great representational capacity for nonlinear functions. In addition, the closed-form expressions for the prediction and the uncertainty estimate in kernel-based models allow the development of algorithms based on nonlinear function approximation that are amenable to theoretical analysis. Kernel-based models also serve as an intermediate step towards understanding the deep learning-based models (see, e.g., Yang et al., 2020) based on the Neural Tangent (NT) kernel approach (Jacot et al., 2018).

38th Conference on Neural Information Processing Systems (NeurIPS 2024).

The infinite-horizon average-reward setting has been extensively explored under the tabular structure (Auer et al., 2008; Wei et al., 2020; Zhang and Xie, 2023). Under the performance measure of *regret*, defined as the difference in the total reward achieved by a learning algorithm over $T$ steps and that of the optimal stationary policy, performance bounds of $\mathcal{O}(\mathrm{poly}(|\mathcal{S}|, |\mathcal{A}|)\sqrt{T})$ have been established (see, e.g., Zhang et al., 2020), where $\mathcal{S}$ and $\mathcal{A}$ represent the state and action spaces, respectively, and the regret grows polynomial with their sizes. It is assumed for these results that the MDP is weakly communicating, a condition necessary for achieving sublinear regret (Bartlett and Tewari, 2009). Averaged over $T$ steps, the regret diminishes as $T$ increases, thereby offering what is known as a *no-regret* performance guarantee. The applicability of the tabular setting is limited, as many real-world problems feature very large or potentially infinite state-action spaces. Consequently, recent literature has explored the use of function approximation in RL, particularly through linear models (Abbasi-Yadkori et al., 2019a,b; Hao et al., 2021; Wei et al., 2021). This approach represents the value function or the transition model via a linear transformation applied to a predefined feature mapping. In the linear setting, regret bounds of $\mathcal{O}((dT)^{\frac{3}{4}})$ have been established (Wei et al., 2021), where $d$ represents the ambient dimension of the linear feature map. Kernel-based models can be considered as linear models in the feature space of the kernel. That, however, is often infinite dimensional ($d = \infty$). As such, the results with linear models do not translate to the kernel-based settings, necessitating novel analytical techniques. Also, for a discussion on further limitations of the linear models, see Lee and Oh (2023).

In this work, we propose the first RL algorithm in the infinite horizon average reward setting with non-linear function approximation using kernel-ridge regression. This is one of the most general models that lends well to theoretical analysis. Our algorithm, referred to as Kernel-based Upper Confidence Bound (KUCB-RL), utilizes kernel ridge regression to build predictor and uncertainty estimates for the expected value function. Inspired by the principle of optimism in the face of uncertainty and equipped with these statistics, KUCB-RL builds an upper confidence bound on the state-action value function over a future window of $w$ steps. This bound serves as a proxy $q_t$, at each step $t$, for the state-action value function over this future window. At each step $t$ with the current state $s_t$, the action is selected greedily with respect to this proxy: $a_t = \arg\max_{a \in \mathcal{A}} q_t(s_t, a)$. This approach resembles the acquisition function based algorithms such as GP-UCB and GP-TS, using Upper Confidence Bound and Thompson sampling, respectively, in the context of kernel-based bandits, also known as Bayesian optimization (Srinivas et al., 2010; Chowdhury and Gopalan, 2017). Kernel-based bandit setting corresponds to the degenerate case of $|\mathcal{S}| = 1$. In comparison, in the RL setting, the action is selected based on the current state, and the reward depends on both the state and the action. A kernel-based model is used to provide predictions for the expected value function, which varies due to the Markovian nature of the temporal dynamics. This makes the RL problem significantly more challenging than the bandit problem where the predictions are derived for a fixed reward function. To address this latter challenge, we derive a novel kernel-based confidence interval that is applicable across RL problems.

## 1.1 Contributions

To summarize, our contributions are as follows. We develop a kernel based optimistic algorithm for the infinite horizon average reward setting, referred to as KUCB-RL. We establish no-regret guarantees for the proposed learning algorithm, which is the first for this setting to the best of our knowledge. Specifically, in Theorem 3, we prove a regret bound of $\mathcal{O}\left(\frac{T}{w} + \left(w + \frac{w}{\sqrt{\rho}}\sqrt{\gamma(T; \rho) + \log(\frac{T}{\delta})}\right)\sqrt{\rho T \gamma(T; \rho) + \rho^2 w^2 \gamma(T; \rho)\gamma(T/w; \rho)}\right)$, at a $1 - \delta$ confidence level, where $\rho$ is the parameter of kernel ridge regression and $\gamma(T; \rho)$ is the maximum information gain, a kernel specific complexity term (see Section 2). This regret bound translates to $\tilde{\mathcal{O}}\left(d^{\frac{1}{2}}T^{\frac{3}{4}}\right)$ in the special case of a linear model, recovering the best existing results (Wei et al., 2021) in dependence on $T$, and improving by a factor of $d^{\frac{1}{4}}$. When applied to very smooth kernels with exponential eigendecay such as the Squared Exponential (SE) kernel, we obtain a regret of $\tilde{\mathcal{O}}(T^{\frac{3}{4}})$, with the notation $\tilde{\mathcal{O}}$ hiding logarithmic factors. For one of the most general cases, the kernels with polynomial eigendecay with parameter $p > 1$ (See Definition 1), that includes, for example, the Matérn family and NT kernels, we show that our regret bound translates to $\tilde{\mathcal{O}}(T^{\frac{3p+5}{4p+4}})$, which constitutes a no-regret guarantee. To highlight the significance of this result, we point out that no-regret guarantees for GP-UCB in the degenerate case of bandits were established only recently

in Whitehouse et al. (2024), while the initial studies of GP-UCB (as well as GP-TS) (Srinivas et al., 2010; Chowdhury and Gopalan, 2017) did not provide no-regret guarantees for the case of polynomial eigendecay. As part of our analysis, in Theorem 1, we develop a novel confidence interval applicable across kernel-based RL problems that contributes to the eventual improved results.

## 1.2 Related Work

The vast RL literature can be categorized across various dimensions. In addition to the average reward, episodic, and discounted settings, as well as tabular, linear, and kernel-based structures mentioned above, other notable distinctions among settings include model-based versus model-free approaches, and offline versus online versus settings where the existence of a generative model is assumed (allowing the learning algorithm to sample the state-action of its choice at each step, rather than following the Markovian trajectory). Covering the entire breadth of RL literature is challenging. Here, we will focus on highlighting and providing comparisons with the most closely related works, particularly in terms of their setting and structure.

The kernel-based MDP structure has been considered in several recent works under the episodic setting (Yang et al., 2020; Vakili and Olkhovskaya, 2023; Chowdhury and Oliveira, 2023; Domingues et al., 2021; Vakili, 2024). The regret bound proven in Yang et al. (2020) for the episodic setting applies only to very smooth kernels such as SE kernel. Vakili and Olkhovskaya (2023) addressed this limitation by extending the results to Matérn and NT families of the kernels, albeit with a sophisticated algorithm that actively partitions the state-action domain into possibly many subdomains, using only the observations within each subdomain to obtain kernel-based prediction and uncertainty estimates. Their work is also based on a particular assumption that relates the kernel eigenvalues to the size of the domain. The work of Chowdhury and Oliveira (2023) is most closely related to ours in terms of kernel-related assumptions. Specifically, our Assumption 4 is identical to Assumption 1 of Chowdhury and Oliveira (2023). They establish a regret bound of $\mathcal{O}(H\gamma(N;\rho)\sqrt{N})$ for the episodic MDP setting, where $N$ is the number of episodes, $\gamma(N;\rho)$ is the maximum information gain, a kernel-related complexity term, $H$ is the episode length and the value of $\rho$ is a fixed constant close to 1. However, their regret bounds do not apply to general families of kernels, such as those with polynomially decaying eigenvalues (see Section 2.2 for the definition) including Matérn and NT kernels, as for this family of kernels $\gamma(N;\rho)$ possibly grows faster than $\sqrt{N}$. As a result, a no-regret guarantee cannot be established in many cases of interest. In comparison, the infinite horizon setting considered in this work is more challenging than the episodic setting as evident when comparing these settings with linear modeling. For this more challenging setting, we establish no-regret guarantees. A key element of our improved results is the novel confidence interval we utilize in our analysis (Theorem 1). This result is general and can be used across RL problems, for example, improving the results of Chowdhury and Oliveira (2023) as well.

In the tabular case, a lower bound of $\Omega(\sqrt{D|\mathcal{S}||\mathcal{A}|T})$ on regret was established in Auer et al. (2008) in the infinite-horizon average-reward setting, where $D$ is the diameter of the MDP. For ergodic MDPs, Wei et al. (2020) shows a regret bound of $\tilde{\mathcal{O}}(\sqrt{t_{\text{mix}}^3|\mathcal{S}||\mathcal{A}|T})$, where $t_{\text{mix}}$ is the mixing time of an ergodic MDP. Furthermore, under the broader assumption of weakly communicating MDPs, which is necessary for low regret (Bartlett and Tewari, 2012), the best existing regret bound of model-free algorithms is $\tilde{\mathcal{O}}(|\mathcal{S}|^5|\mathcal{A}|^2\sqrt{T})$, achieved by the recent work of Zhang and Xie (2023). Several works have studied linear function approximation in the infinite horizon average reward setting under strong assumptions of uniformly mixing and uniformly excited feature conditions (Abbasi-Yadkori et al., 2019a,b; Hao et al., 2021). Notably, Hao et al. (2021) achieved a regret bound of $\tilde{\mathcal{O}}\left(\frac{1}{\sigma}\sqrt{t_{\text{mix}}^3 T}\right)$ under the linear bias function assumption, where $\sigma$ is the smallest eigenvalue of policy-weighted covariance matrix. Under the much less restrictive setting of Bellman optimality equation assumption (Assumption 1) for linear MDP, Wei et al. (2021) provides an algorithm with regret guarantee of $\tilde{\mathcal{O}}((dT)^{3/4})$. We also consider our kernel-based approach under this general assumption on MDP. Furthermore, for examples of infeasible algorithms in the literature, see Wei et al. (2021), Algorithm 1. There also exists a separate model-based approach to the problem where the transition probability distribution (model) is learned and used for planning, usually requiring high memory and computational complexity and utilizing substantially different techniques and assumptions. While this approach is studied under tabular settings (Bartlett and Tewari, 2009; Auer et al., 2008) and linear settings (Wu et al., 2022), it is not clear whether model-based approaches can

be feasibly constructed in the kernel-based setting, due to the space complexity of a kernel-based model.

Our work is also related to the simpler problem of kernelized bandits (Srinivas et al., 2010; Chowdhury and Gopalan, 2017; Vakili et al., 2021b; Li and Scarlett, 2022; Salgia et al., 2021). Our construction of the confidence interval for the RL setting has been inspired by the previous work on bandits, utilizing novel analysis introduced in Whitehouse et al. (2024). Bandit settings can be considered a degenerate case of the RL framework with $|\mathcal{S}| = 1$. In comparison, the temporal dependencies of MDP introduce substantial challenges, and the confidence intervals used in the bandit setting cannot be directly applied.

We summarize the most closely related work with a focus on model-free feasible algorithms in Table 1. We present the existing regret bounds under various assumptions on MDP and its structure (tabular, linear, kernel-based). The assumptions include weakly communicating MDP (See Puterman, 1990, Section 8.3.1), Bellman optimality equation (our Assumption 1), and uniform mixing assumption (see Wei et al., 2021, Assumption 3). For a formal definition of linear MDP, see Wei et al. (2021), Assumption 2, and for the linear bias function case, see Wei et al. (2021), Assumption 4.

Table 1: Summary of the existing regret bounds in the infinite horizon average reward setting under various cases with respect to MDP structure (tabular, linear, kernel based) and assumptions.

| Algorithm | Regret | MDP Assumption | Structure |
| --- | --- | --- | --- |
| UCB-AVG (Zhang and Xie, 2023) | $\tilde{\mathcal{O}}(|\mathcal{S}|^5|\mathcal{A}|^2\sqrt{T})$ | Weakly Communicating | Tabular |
| OLSVI.FH (Wei et al., 2021) | $\tilde{\mathcal{O}}((dT)^{3/4})$ | Bellman Optimality Eq. | Linear |
| MDP-Exp2 (Wei et al., 2021) | $\tilde{\mathcal{O}}(\sqrt{t_{\text{mix}}^3|\mathcal{S}||\mathcal{A}|T})$ | Uniform Mixing | Linear Bias function |
| **KUCB-RL** (Algorithm 1) | $\tilde{\mathcal{O}}\left(d^{\frac{1}{2}}T^{\frac{3}{4}}\right)$ | Bellman Optimality Eq. | Linear |
| **KUCB-RL** (Algorithm 1) | $\tilde{\mathcal{O}}\left(T^{\frac{3}{4}}\right)$ | Bellman Optimality Eq. | Kernel-based (exponential) |
| **KUCB-RL** (Algorithm 1) | $\tilde{\mathcal{O}}\left(T^{\frac{3p+5}{4p+4}}\right)$ | Bellman Optimality Eq. | Kernel-based (polynomial) |

## 2 Problem Formulation

In this section, we overview the background on infinite horizon average reward (undiscounted) MDPs and kernel based modelling.

### 2.1 Infinite Horizon Average Reward MDP

An undiscounted MDP is described by the tuple $(\mathcal{S}, \mathcal{A}, r, P)$ where $\mathcal{S}$ is a state space with a possibly infinite number of elements, $\mathcal{A}$ is a finite action set, $r : \mathcal{S} \times \mathcal{A} \to [0, 1]$ is the reward function, and $P(\cdot|s, a)$ is the unknown transition probability distribution over $\mathcal{S}$ of the next state when action $a$ is selected at state $s$. Throughout the paper we use the notation $z = (s, a)$ for the state-action pairs, and $\mathcal{Z} = \mathcal{S} \times \mathcal{A}$.

The learner interacts with the MDP through $T$ steps, starting from an arbitrary initial state $s_1 \in \mathcal{S}$. At each step $t$, the learner observes state $s_t$ and takes an action $a_t$ resulting in a reward $r(s_t, a_t)$. The next state $s_{t+1}$ is revealed as a sample drawn from the transition probability distribution: $s_{t+1} \sim P(\cdot|s_t, a_t)$.

The goal of the learner is to compete against any fixed stationary policy. A stationary policy $\pi : \mathcal{S} \to \mathcal{A}$ is a possibly random mapping from the states to actions. The long-term average reward of a stationary policy $\pi$, starting from state $s \in \mathcal{S}$, is defined as:

$$J^\pi(s) = \liminf_{T \to \infty} \frac{1}{T}\mathbb{E}\left[\sum_{t=1}^{T} r(s_t, a_t) \,\middle|\, s_1 = s, \forall t \geq 1, a_t = \pi(s_t), s_{t+1} \sim P(\cdot|s_t, a_t)\right].$$

We assume that the MDP belongs to the broad class of MDPs where the following form of the Bellman optimality equation holds:

**Assumption 1** (Bellman optimality equation). *There exists $J^\star \in \mathbb{R}$ and bounded measurable functions $v^\star : \mathcal{S} \to \mathbb{R}$ and $q^\star : \mathcal{S} \times \mathcal{A} \to \mathbb{R}$ such that the following conditions are satisfied for all*

*states $s \in \mathcal{S}$ and actions $a \in \mathcal{A}$ :*

$$J^\star + q^\star(s,a) = r(x,a) + \mathbb{E}_{s' \sim P(\cdot|s,a)} \left[ v^\star(s') \right], \quad v^\star(s) = \max_{a \in \mathcal{A}} q^\star(s,a). \tag{1}$$

This assumption was also used for the linear MDP case in Wei et al. (2021). By applying the Bellman optimality equation, it can be shown that a policy $\pi^\star(s) = \arg\max_{a \in \mathcal{A}} q^\star(s,a)$, which deterministically selects actions that maximize $q^\star$ in the current state, is the optimal policy $\pi^\star = \arg\max_\pi J^\pi$, with $J^{\pi^\star}(s) = J^\star$, for all $s$ (Wei et al., 2021).

For the finite state setting, Assumption 1 follows from the weakly communicating MDP assumption (see, e.g., Puterman, 1990, Chapter 9). Assumption 1 also holds under several other common conditions (Hernández-Lerma (2012), Section 3.3).

The learner's performance is measured by *regret*, which is defined as the difference in total reward between the learner and the optimal stationary policy. Specifically,

$$\mathcal{R}(T) = \sum_{t=1}^{T} (J^\star - r(s_t, a_t)). \tag{2}$$

We emphasize that under Assumption 1, for any initial state $s_1 \in \mathcal{S}$, $J^{\pi^\star}(s_1) = J^\star$, that is reflected in our regret definition.

For any value function $v : \mathcal{S} \to \mathbb{R}$, throughout the paper, we use the notation

$$[Pv](z) = \mathbb{E}_{s' \sim P(\cdot|z)}[v(s')]$$

for the expected value function of the next state.

## 2.2 Kernel-Based Models and the RKHS

Consider a positive definite kernel $k : \mathcal{Z} \times \mathcal{Z} \to \mathbb{R}$. Let $\mathcal{H}_k$ be the reproducing kernel Hilbert space (RKHS) induced by $k$, where $\mathcal{H}_k$ contains a family of functions defined on $\mathcal{Z}$. Let $\langle \cdot, \cdot \rangle_{\mathcal{H}_k} : \mathcal{H}_k \times \mathcal{H}_k \to \mathbb{R}$ and $\| \cdot \|_{\mathcal{H}_k} : \mathcal{H}_k \to \mathbb{R}$ denote the inner product and the norm of $\mathcal{H}_k$, respectively. The reproducing property implies that for all $f \in \mathcal{H}_k$, and $z \in \mathcal{Z}$, $\langle f, k(\cdot, z) \rangle_{\mathcal{H}_k} = f(z)$. Mercer theorem implies that $k$ can be represented using a possibly infinite dimensional feature map:

$$k(z, z') = \sum_{m=1}^{\infty} \lambda_m \varphi_m(z) \varphi_m(z'), \tag{3}$$

where $\lambda_m > 0$, and $\sqrt{\lambda_m} \varphi_m \in \mathcal{H}_k$ form an orthonormal basis of $\mathcal{H}_k$. In particular, any $f \in \mathcal{H}_k$ can be represented using this basis and weights $w_m \in \mathbb{R}$ as

$$f = \sum_{m=1}^{\infty} w_m \sqrt{\lambda_m} \varphi_m,$$

where $\|f\|_{\mathcal{H}_k}^2 = \sum_{m=1}^{\infty} w_m^2$. A formal statement and the details are provided in Appendix 8. We refer to $\lambda_m$ and $\varphi_m$ as (Mercer) eigenvalues and eigenfunctions of kernel $k$, respectively.

## 2.3 Kernel-Based Prediction

Kernel-based models provide powerful predictors and uncertainty estimators which can be leveraged to guide the RL algorithm. In particular, consider a fixed unknown function $f \in \mathcal{H}_k$. Assume a $t \times 1$ vector of noisy observations $\boldsymbol{y}_t = [y_i = f(z_i) + \varepsilon_i]_{i=1}^{t}$ at observation points $\{z_i\}_{i=1}^{t}$ is provided, where $\varepsilon_i$ are independent zero mean noise terms. Kernel ridge regression provides the following predictor and uncertainty estimate, respectively (see, e.g., Schölkopf et al., 2002),

$$\hat{f}_t(z) = k_t^\top(z)(K_t + \rho I)^{-1} \boldsymbol{y}_t,$$
$$\sigma_t^2(z) = k(z,z) - k_t^\top(z)(K_t + \rho I)^{-1} k_t(z), \tag{4}$$

where $k_t(z) = [k(z, z_1), \ldots, k(z, z_t)]^\top$ is a $t \times 1$ vector of the kernel values between $z$ and observations, $K_t = [k(z_i, z_j)]_{i,j=1}^{t}$ is the $t \times t$ kernel matrix, $I$ is the identity matrix appropriately sized to match $K_t$, and $\rho > 0$ is a free regularization parameter.

Confidence bounds of the form $|f(z) - \hat{f}_t(z)| \le \beta(\delta)\sigma_t(z)$ are established, for a confidence interval width multiplier $\beta(\delta)$ at a confidence level $1 - \delta$, which depends on the assumptions on the setting and the noise. We will establish a such confidence interval specific to the RL setting, in Theorem 1, and utilize it in our regret analysis.

## 2.4 Kernel-Based Modelling in RL

In our RL setting, we use a kernel-based model to predict the expected value function. In particular, for a given transition probability distribution $P(s'|\cdot, \cdot)$ and a value function $v : \mathcal{S} \to \mathbb{R}$, we define $f = [Pv]$ and use past observations to form predictions and uncertainty estimates for $f$, as detailed in the following section. The value functions vary due to the Markovian nature of the temporal dynamics. To effectively use the confidence intervals established by the kernel-based models on $f$, we require the following assumption.

**Assumption 2.** *We assume $P(s'|\cdot, \cdot) \in \mathcal{H}_k$, for some positive definite kernel $k$, and $\|P(s'|\cdot, \cdot)\|_{\mathcal{H}_k} \le 1$ for all $s' \in \mathcal{S}$.*

## 2.5 Eigendecay and Information Gain

Our regret bounds are presented in terms of maximum information gain which is a kernel-specific complexity term. Specifically, for a kernel $k$ and sets of observation points $\{z_i\}_{i=1}^t$, we define the maximum information gain $\gamma(t; \rho)$ as follows

$$\gamma(t; \rho) = \sup_{\{z_i\}_{i=1}^t \subset \mathcal{Z}} \frac{1}{2} \log \det\left(I + \frac{K_t}{\rho}\right),$$

where $\rho > 0$, and $K_t$ is the kernel matrix defined in Section 2.3. Several works have established upper bounds on $\gamma(t; \rho)$. In the special case of a $d$-dimensional linear kernel, we have $\gamma(t; \rho) = \mathcal{O}(d \log(t))$. For kernels with exponential eigendecay, including SE, $\gamma(t; \rho) = \mathcal{O}(\text{polylog}(t))$ (Srinivas et al., 2010; Vakili et al., 2021b). For kernels with polynomial eigendecay, which represent a crucial case due to challenges in establishing no-regret guarantees in RL and bandits, and include kernels of both practical and theoretical interest such as the Matérn family and NT kernels, we first provide the definition below and then the bound on $\gamma$.

**Definition 1.** *A kernel $k$ is said to have a $p$-polynomial eigendecay if $\forall m \ge 1$, $\lambda_m \le C m^{-p}$, for some $p > 1$, $C > 0$ where $\lambda_m$ are the Mercer eigenvalues of the kernel in decreasing order.*

For kernels with $p$-polynomial eigendecay, we have (Vakili et al., 2021b, Corollary 1):

$$\gamma(t; \rho) = \mathcal{O}\left(\left(\frac{t}{\rho}\right)^{\frac{1}{p}} \left(\log\left(1 + \frac{t}{\rho}\right)\right)^{1 - \frac{1}{p}}\right).$$

# 3 KUCB-RL Algorithm

In this section, we introduce our algorithm, Kernel-based Upper Confidence Bound for Reinforcement Learning (KUCB-RL). The algorithm's structure is similar to acquisition-based kernel bandit algorithms such as GP-UCB (Srinivas et al., 2010), where each action is chosen as the maximizer of an acquisition function. We construct an optimistic proxy $q_t$ for the state-action value function. At each step $t$, given the current state $s_t$, the action $a_t$ is selected as the maximizer of $q_t(s_t, a)$ over $a$. This proxy $q_t$ is derived using past observations of transitions, employing kernel ridge regression to provide a prediction and uncertainty estimate for the state-action value function over a future window of size $w \in \mathbb{N}$. The proxy is established as an upper confidence bound, following the principle of optimism in the face of uncertainty. The value functions are computed in batches of $w$ steps, and the derived policies are unrolled over the subsequent $w$ steps. The details are presented next.

We define a fixed window size, $w \in \mathbb{N}$, which represents the future interval that the algorithm will consider. For a given $t_0$ where $(t_0 \mod w) = 0$, including $t_0 = 0$, we initialize $v_{t_0+w+1}(s) = 0, \forall s \in \mathcal{S}$, reflecting the algorithm's consideration of the reward within this future window of size $w$. Subsequently, we recursively obtain proxies $q_t$ and $v_t$ for all steps $t \in \{t : t_0 + 1 \le t \le t_0 + w\}$. Let $f_t$ denote $[Pv_{t+1}]$, $\hat{f}_t$ represent the kernel ridge predictor of $[Pv_{t+1}]$, and $\sigma_t$ be its uncertainty

**Algorithm 1** Kernel-based Upper Confidence Bound for Reinforcement Learning (KUCB-RL)

---

**Require:** Regularization parameter $\rho$, window size $w$, confidence interval width multiplier $\beta$, confidence level $1 - \delta$, $\mathcal{S}, \mathcal{A}, r$.
1: **for** $t = 0, 1, 2, \cdots$ **do**
2:     **if** $(t \mod w) = 0$ **then**
3:         Let $v_{t+w+1} = \mathbf{0}$;
4:         **for** $h = 1, 2, \cdots, w$ **do**
5:             Compute $q_{t+w+1-h}$ and $v_{t+w+1-h}$ using equations (6) and (7).
6:         **end for**
7:     **end if**
8:     Select $a_t = \arg\max_{a \in \mathcal{A}} q_t(s_t, a)$;    Observe $s_{t+1} \sim P(\cdot | s_t, a_t)$ and receive $r(s_t, a_t)$.
9: **end for**

---

estimator. The predictor and the uncertainty estimator are derived using the data set $\mathcal{D}_{t_0}$, which contains observations of past transitions up to $t_0$. We use the notation $\mathcal{D}_t = \{(s_j, a_j, s_{j+1})\}_{j \leq t}$ for the past transitions, and also define $\boldsymbol{v}_{t+1, t_0} = [v_{t+1}(s_2), v_{t+1}(s_3), \cdots, v_{t+1}(s_{t_0+1})]^\top$, for the values of the proxy value function at the history of state observations. We then have

$$\hat{f}_t(z) = k_{t_0}^\top(z) \left(K_{t_0} + \rho I\right)^{-1} \boldsymbol{v}_{t+1, t_0},$$
$$\sigma_t^2(z) = k(z, z) - k_{t_0}^\top(z) \left(K_{t_0} + \rho I\right)^{-1} k_{t_0}(z), \tag{5}$$

where $k_t(z) = [k(z, z_1), k(z, z_2), \cdots, k(z, z_t))]^\top$ denotes the vector of kernel values between $z$ and $(z_j = (s_j, a_j))_{j \leq t}$ in the history of observations, and $K_t = [k(z_i, z_j)]_{i,j=1}^t$ denotes the kernel matrix.

Equipped with the kernel ridge predictor and uncertainty estimator, we define $q_t$ as an upper confidence bound for $f_t$, as follows:

$$q_t(z) = \Pi_{[0, w]} \left(r(z) + \hat{f}_t(z) + \beta(\delta)\sigma_t(z)\right), \quad \forall z \in \mathcal{Z}, \tag{6}$$

where $1 - \delta$ represents a confidence level, and $\beta(\delta)$ is a confidence interval width multiplier; the specific value of which is given in Theorem 3. The notation $\Pi_{[a,b]}(\cdot)$ is used for projection on $[a, b]$ interval. This step is natural since with the assumption $r : \mathcal{Z} \to [0, 1]$ the value over a window of size $w$ can not be more than $w$. We also define

$$v_t(s) = \max_{a \in \mathcal{A}} q_t(s, a), \quad \forall s \in \mathcal{S}. \tag{7}$$

By iteratively updating from $t = t_0 + w$ to $t = t_0 + 1$, we compute the values of $q_t$ and $v_t$ for all $t$ from $t_0 + 1$ to $t_0 + w$. Then, we unroll the learned policy over the subsequent $w$ steps, as the greedy policy with respect to $q_t$:

$$a_t = \arg\max_{a \in \mathcal{A}} q_t(s_t, a). \tag{8}$$

A pseudocode is provided in Algorithm 1.

**Computational Complexity.** KUCB-RL enjoys a polynomial computational complexity of $\mathcal{O}(\frac{T^4}{w})$, where the bottleneck is the matrix inversion step in (5) in kernel ridge regression every $w$ steps. This is not unique to our work and is common across kernel-based supervised learning, bandit, and RL literature. Luckily, sparse approximation methods such as Sparse Variational Gaussian Processes (SVGP) or the Nyström method significantly reduce the computational complexity of matrix inversion step (to as low as linear in some cases), while maintaining the kernel-based confidence intervals and, consequently, the eventual rates (see, e.g., Vakili et al., 2022, and references therein). These results are, however, generally applicable and not specific to our problem.

## 4   Regret Bounds for KUCB-RL

In this section, we provide analytical results on the performance of KUCB-RL. We prove the first sublinear regret bounds in undiscounted RL setting under general assumptions based on kernel-based modelling. We first derive a novel confidence interval that is broadly applicable to the kernel-based RL problems. We then utilize this result to establish bounds on the regret of KUCB-RL.

## 4.1 Confidence Intervals for Kernel Based RL

The analysis of our algorithm utilizes confidence intervals of the form $|f_t(z) - \hat{f}_t(z)| \le \beta(\delta)\sigma_t(z)$, where $f_t = [Pv_t]$ denotes the expected value of a value function $v_t$, and $\hat{f}_t$ and $\sigma_t$ represent the kernel ridge predictor and the uncertainty estimate of $f_t$. Here, $\beta(\delta)$ represents the width multiplier for the confidence interval at a $1 - \delta$ confidence level. Similar confidence intervals are established in kernel ridge regression for a fixed function $f$ in the RKHS of a specified kernel $k$ (see, e.g., Abbasi-Yadkori, 2013; Srinivas et al., 2010; Chowdhury and Gopalan, 2017; Vakili et al., 2021a; Whitehouse et al., 2024). In the RL context, specific considerations are required as both $f_t = [Pv_t]$ and the observation noise depend on the value function $v_t$ that varies due to the Markovian nature of the temporal dynamics. We note that in this setting, for a given value function $v : \mathcal{S} \to \mathbb{R}$, the observation noise is captured by $v(s_{t+1}) - [Pv](s_t, a_t)$. A possible approach involves deriving confidence intervals that apply to a class $\mathcal{V}$ of value functions. Such results appear in some of the existing work (Chowdhury and Oliveira, 2023; Vakili and Olkhovskaya, 2023). The result most closely related to our is Chowdhury and Oliveira (2023), which derives its confidence interval under the exact same kernel related assumptions as our work, but for the episodic MDP setting. With the same assumptions, the confidence interval that we establish is different from the one in Chowdhury and Oliveira (2023). In particular, their confidence interval is applicable to a specific value of kernel ridge regression parameter $\rho$, constrained by their proof techniques. Inspired by Whitehouse et al. (2024), which established a confidence interval for kernel ridge regression (not within the RL context) but allowed for a judicious selection of $\rho$, we prove a new confidence interval suitable for the RL setting that allows tuning parameter $\rho$. As a result, we obtain the first improved no-regret algorithms in this setting.

**Theorem 1** (Confidence Bound). *Consider $v : \mathcal{S} \to \mathbb{R}$, a conditional probability distribution $P(s|z)$, $s \in \mathcal{S}$, $z \in \mathcal{Z}$, and two positive definite kernels $k : \mathcal{Z} \times \mathcal{Z} \to \mathbb{R}$ and $k' : \mathcal{S} \times \mathcal{S} \to \mathbb{R}$, where $\mathcal{Z} = \mathcal{S} \times \mathcal{A}$ is compact subset of $\mathbb{R}^d$. Let $f = [Pv]$ and assume $\|v\|_{\mathcal{H}_{k'}} \le C_v$, $v(s) \le w, \forall s \in \mathcal{S}$, and $\|f\|_{\mathcal{H}_k} \le C_f$, for some $C_v, w, C_f > 0$. A dataset $\{(z_i, s'_i)\}_{i=1}^n \subset (\mathcal{Z} \times \mathcal{S})^n$ is provided such that $s'_i \sim P(\cdot|z^i)$. Let $\lambda_m$, $m = 1, 2, \cdots$ denote the Mercer's eigenvalues of $k'$ in a decreasing order and $\psi_m$ denote the corresponding eigenfunctions, with $\psi_m \le \psi_{\max}$ for some $\psi_{\max} > 0$.*

*Let $\hat{f}_n$ and $\sigma_n$ be the kernel ridge predictor and the uncertainty estimate of $f$ using the observations:*

$$\hat{f}_n(z) = k_n^\top(z)(\rho I + K_n)^{-1}\boldsymbol{v}_n, \quad \sigma_n^2(z) = k(z, z) - k_n^\top(z)(\rho I + K_n)^{-1}k_n(z),$$

*where $\boldsymbol{v}_n = [v(s'_1), v(s'_2), \cdots, v(s'_n)]^\top$ is the vector of observations.*

*For all $z \in \mathcal{Z}$ and $v : \|v\|_{\mathcal{H}_{k'}} \le C_v$, the following holds, with probability at least $1 - \delta$,*

$$|f(z) - \hat{f}_n(z)| \le \beta(\delta)\sigma_n(z),$$

*with $\beta(\delta) =$*

$$C_f + \frac{C_v\psi_{\max}}{\sqrt{\rho}}\left(\sum_{m=1}^M \lambda_m\right)^{\frac{1}{2}}\left(2\log\left(\sqrt{\frac{M}{\delta}}det(I + \rho^{-1}K_n)\right)\right)^{\frac{1}{2}} + \frac{2C_v\psi_{\max}}{\sqrt{\rho}}\left(n\sum_{m=M+1}^\infty \lambda_m\right)^{\frac{1}{2}}.$$

We can simplify the presentation of $\beta$ under the following assumption.

**Assumption 3.** *For the kernel $k'$, we assume that for some $C_1, C_2$ and $q > 0$, $\sum_{m=1}^M \lambda_m \le C_1$ and, $\sum_{m=M+1}^\infty \lambda_m \le C_2 M^{-q}$ for any $M \in \mathbb{N}$.*

This is a mild assumption. For example, a $p$-polynomial eigendecay profile with $p > 1$, which applies to a large class of common kernels including SE, Matérn and NT kernels, satisfies this assumption with $C_1 = \frac{pC}{p-1}$, $C_2 = \frac{C}{p-1}$, and $q = p - 1$, where $C$ is the constant specified in Definition 1.

**Remark 2.** *Under Assumption 3, the expression of $\beta$ in Theorem 3 can be simplified as*

$$\beta(\delta) = \mathcal{O}\left(C_f + \frac{C_v}{\sqrt{\rho}}\sqrt{\log(\frac{n}{\delta}) + \gamma(\rho; n)}\right).$$

Remark 2 can be observed by selecting $M = \lceil n^{\frac{1}{q}} \rceil$ in the expression of $\beta(\delta)$, which provides a straightforward presentation of the confidence interval width multiplier.

The proof of Theorem 1 involves the Mercer representation of $v$ in terms of $\psi_m$. The expression of the prediction error $|f(z) - \hat{f}_n(z)|$ is then decomposed into error terms corresponding to each $\psi_m$. We then partition these terms into the first $M$ elements corresponding to eigenfunctions with the largest $M$ eigenvalues and the rest. For each of the first $M$ eigenfunctions, we obtain high probability bounds using existing confidence intervals from Whitehouse et al. (2024). Summing up over $m$, and using a bound based on uncertainty estimates, we achieve a high probability bound—corresponding to the second term in the expression of $\beta(\delta)$. We then bound the remaining $m > M$ elements based on the decay of Mercer eigenvalues—corresponding to the third term in the expression of $\beta(\delta)$. A detailed proof is provided in Appendix 6.

Theorem 1 is presented in a self-contained way, making it broadly applicable across various RL settings. In the following section, we apply this theorem within the analysis of the infinite horizon average reward setting to obtain a no-regret algorithm. This is the first no-regret algorithm within this setting under general kernel-related assumptions.

## 4.2 Regret Analysis of KUCB-RL

The weakest assumption regarding value functions is realizability, which suggests that the optimal value function $v^\star$ either belong to the an RKHS or are at least well-approximated by its elements. In the degenerate case of bandits with $|\mathcal{S}| = 1$, realizability alone is sufficient for provably efficient algorithms (Srinivas et al., 2010; Chowdhury and Gopalan, 2017; Vakili et al., 2021a). However, for general MDPs, realizability is inadequate, necessitating stronger assumptions (Jin et al., 2020; Wang et al., 2019; Chowdhury and Oliveira, 2023). Building on these works, our main assumption involves a closure property for all value functions within the following class:

$$\mathcal{V} = \left\{ s \to \min \left\{ w, \max_{a \in \mathcal{A}} \left\{ r(s,a) + \boldsymbol{\phi}^\top(s,a)\boldsymbol{\theta} + \beta\sqrt{\boldsymbol{\phi}^\top(s,a)\Sigma^{-1}\boldsymbol{\phi}(s,a)} \right\} \right\} \right\}, \quad (9)$$

where $\beta \in \mathbb{R}$ and $\beta > 0$, $\|\boldsymbol{\theta}\| < \infty$, and $\Sigma$ is an $\infty \times \infty$ matrix operator such that $\Sigma \succeq \rho I$ for some $\rho > 0$, and $\boldsymbol{\phi} = [\phi_1, \phi_2, \cdots]$, where $\phi_m = \sqrt{\lambda_m}\varphi_m$, and $\lambda_m$ and $\varphi_m$ are the Mercer eigenvalues and eigenfunctions corresponding to a kernel $k$ defined on $\mathcal{Z} \times \mathcal{Z}$. We assume $\mathcal{V}$ is a subset of the RKHS of a kernel $k'$ defined on $\mathcal{S} \times \mathcal{S}$.

**Assumption 4** (Optimistic Closure). *For any $v \in \mathcal{V}$, and for some positive constant $C_v$, we have $\|v\|_{k'} \leq C_v$. Additionally, for $v : \mathcal{S} \to [0, w]$, we assume $C_v = \mathcal{O}(w)$.*

This technical assumption is the same as Assumption 1 in Chowdhury and Oliveira (2023). The optimistic closure assumption in the kernel-based setting is strictly weaker than the ones explored in the context of generalized linear function approximation (Wang et al., 2020).

**Theorem 3.** *Consider the undiscounted MDP setting described in Section 2. Run KUCB-RL given in Algorithm 1 for $T$ steps. Under Assumptions 1, 2, 3, and 4, the regret of KUCB-RL, defined in Equation (2), satisfies, with probability at least $1 - \delta$*

$$\mathcal{R}(T) = \mathcal{O}\left( \frac{T}{w} + \left( w + \frac{w}{\sqrt{\rho}}\sqrt{\gamma(T;\rho) + \log\left(\frac{T}{\delta}\right)} \right) \sqrt{\rho T \gamma(T;\rho) + \rho^2 w^2 \gamma(T;\rho)\gamma(T/w;\rho)} \right).$$

The proof of Theorem 3 utilizes standard methods from the analysis of optimistic algorithms in RL and bandits, such as the elliptical potential lemma, leverages the confidence interval proven in Theorem 1, and also incorporates novel techniques. Algorithm 1 updates the observation set every $w$ steps, requiring us to characterize and bound the effect of this delay in the proof. A straightforward application of the elliptical potential lemma results in loose bounds that do not guarantee no-regret. In Lemma 4, we establish a tight bound on the sum of standard deviations of a sequence of points with delayed updates of the observation sets, contributing to the improved regret bounds. This is independently a useful result in other settings with delayed updates, such as delayed feedback settings (Vakili et al., 2023a; Kuang et al., 2023) or when observations are provided in a batch (Chowdhury and Gopalan, 2019). The details are provided in Appendix 7.

There is an apparent trade-off in choosing the window size. Intuitively, this trade-off balances the strength of the value function against the strength of the noise. A larger $w$ is preferred to capture the long-term performance of the policy, but a larger $w$ also increases the observation noise affecting

the prediction error in kernel ridge regression. The optimal window size results from an interplay between these two factors, which is reflected in the regret bound.

We next instantiate our regret bounds for some special cases. In the linear case, with a choice of $w = T^{\frac{1}{4}} d^{\frac{-1}{4}}$ and replacing the bound on $\gamma(T; \rho)$, we obtain $\mathcal{R}(T) = \tilde{\mathcal{O}}(d^{\frac{1}{2}} T^{\frac{3}{4}})$, recovering the existing results in their dependence on $T$ and improving by a factor of $d^{\frac{1}{4}}$. For kernels with exponential eigendecay, with a choice of $w = T^{\frac{1}{4}}$ and replacing the bound on $\gamma(T; \rho)$, we obtain $\mathcal{R}(T) = \tilde{\mathcal{O}}(T^{\frac{3}{4}})$. We formalize the result with $p$-polynomial kernels in the following remark as it may be of broader interest.

**Remark 4.** *Under the setting of Theorem 3, with a $p$-polynomial kernel, with the choice of parameters, $w = T^{\frac{p-1}{4p+4}}$ and $\rho = T^{\frac{1}{p+1}}$, we obtain the following no-regret guarantee $\mathcal{R}(T) = \tilde{\mathcal{O}}(T^{\frac{3p+5}{4p+4}})$.*

In the case of a Matérn kernel with smoothness parameter $\nu$, where $p = 1 + \frac{2\nu}{d}$, the regret bound translates to $\mathcal{R}(T) = \mathcal{O}\left(T^{\frac{3\nu+4d}{4\nu+4d}}\right)$. This also directly extends to NT kernels using the equivalence between the RKHS of Matérn kernels and NT kernels with the appropriate smoothness (Vakili et al., 2023b).

## 5 Discussion and Limitations

We proposed KUCB-RL in the infinite horizon average reward setting and proved no-regret guarantees with general kernels, including those with polynomial eigendecay such as Matérn and NT kernels. To highlight the significance of our results, we note that in the case of episodic MDPs, the existing work of (Yang et al., 2020; Chowdhury and Oliveira, 2023) do not provide no-regret guarantees with general kernels. The work of Vakili and Olkhovskaya (2023) utilizes sophisticated domain partitioning techniques and relies on a specific assumption about the scaling of kernel eigenvalues with the size of the domain. We achieve improved rates on regret leveraging a confidence interval proven in Theorem 1, which is applicable across various RL problems. We next point out two main limitations of our work.

Regarding optimality, we can juxtapose our results with the $\Omega(T^{\frac{\nu+d}{2\nu+d}})$ lower bounds proven in (Scarlett et al., 2017), for the degenerate case of bandits with Matérn kernel. Sophisticated algorithms, such as the *sup* variation of optimistic algorithms and those based on sample or domain partitioning (Valko et al., 2013; Salgia et al., 2021; Li and Scarlett, 2022), achieve this lower bound up to logarithmic factors in the case of bandits. However, a no-regret $\tilde{\mathcal{O}}(T^{\frac{\nu+2d}{2\nu+2d}})$ guarantee, though suboptimal, for standard acquisition-based algorithms like GP-UCB has been provided only recently (Whitehouse et al., 2024). While we offer the first no-regret $\tilde{\mathcal{O}}(T^{\frac{3\nu+4d}{4\nu+4d}})$ guarantee in the much more complex setting of RL, we cannot determine whether our results are improvable. This remains an area for future investigation.

Although RKHS elements of common kernels can approximate almost all continuous functions on compact subsets of $\mathbb{R}^d$ (Srinivas et al., 2010), the optimistic closure assumption is somewhat limiting. A rigorous approach involves relaxing this assumption and finding an RKHS element that serves as an upper confidence bound on a function of interest $f$ within the same RKHS. While this method appears to reasonably address the assumption, it is a technically involved problem that invites further contributions from researchers in the field.

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

# 6  Proof of Theorem 1

In this section, we provide a detailed proof the confidence bound given in Theorem 1.

Let us use the notation

$$\alpha_n(z) = k_n^\top(z)(\rho I + K_n)^{-1}, \tag{10}$$

and $\varepsilon_i = v(s_i') - f(z_i)$, $\boldsymbol{\varepsilon}_n = [\varepsilon_1, \varepsilon_2, \cdots, \varepsilon_n]^\top$, $\boldsymbol{f}_n = [f(z_1), f(z_2), \cdots, f(z_n)]^\top$.

This allows us to rewrite the prediction error as

$$\begin{aligned}
f(z) - \hat{f}_n(z) &= f(z) - \alpha_n^\top(z)\boldsymbol{v}_n \\
&= f(z) - \alpha_n^\top(z)(\boldsymbol{f}_n + \boldsymbol{\varepsilon}_n) \\
&= \big(f(z) - \alpha_n^\top(z)\boldsymbol{f}_n\big) - \alpha_n^\top(z)\boldsymbol{\varepsilon}_n.
\end{aligned} \tag{11}$$

The first term on the right-hand side represents the prediction error from noise-free observations, and the second term is the prediction error due to noise. The first term is deterministic (not random) and can be bounded following the standard approaches in kernel-based models, for example using the following result from Vakili et al. (2021a):

**Lemma 1** (Proposition 1 in Vakili et al. (2021a)). *We have*

$$\sigma_n^2(z) = \sup_{f:\|f\|_{\mathcal{H}} \leq 1} (f(z) - \alpha_n^\top(z)f_n)^2 + \rho\,\|\alpha_n(z)\|_{\ell^2}^2\,.$$

Based on this lemma, the first term on the right hand side of (11) can be deterministically bounded by $C_f\sigma_n(z)$ :

$$|f(z) - \alpha_n^\top(z)\boldsymbol{f}_n| \leq C_f\sigma_n(z).$$

The challenging part in Equation (11) is the second term, which is the noise-dependent term $\alpha_n^\top(z)\boldsymbol{\varepsilon}_n$. Next, we provide a high probability bound on this term.

We leverage the Mercer representation of $v$ and write:

$$v(s) = \sum_{m=1}^{\infty} w_m \lambda_m^{\frac{1}{2}} \psi_m(s).$$

We rewrite the observation vector $\boldsymbol{v}_n$ as the sum of a noise term and the noise-free part $f$:

$$v(s_i') = \underbrace{(v(s_i') - f(z_i))}_{\text{Observation noise}} + \underbrace{f(z_i)}_{\text{Noise-free observation}}$$

Using the notation $\overline{\psi}_m(z) = \mathbb{E}_{s' \sim P(\cdot|z)}\psi(s')$, we can rewrite $f(z_i)$ as follows:

$$\begin{aligned}
f(z_i) &= \mathbb{E}_{s \sim P(\cdot|z_i')}[v(s)] \\
&= \mathbb{E}_{s \sim P(\cdot|z_i)}\left[\sum_{m=1}^{\infty} w_m \lambda_m^{\frac{1}{2}} \psi_m(s)\right] \\
&= \sum_{m=1}^{\infty} w_m \lambda_m^{\frac{1}{2}} \mathbb{E}_{s' \sim P(\cdot|z_i)}[\psi_m(s')] \\
&= \sum_{m=1}^{\infty} w_m \lambda_m^{\frac{1}{2}} \overline{\psi}_m(z_i). \tag{12}
\end{aligned}$$

Using this representation, we can rewrite the second term of (11) as follows

$$\sum_{i=1}^{n} \alpha_i(z)\varepsilon_i = \sum_{i=1}^{n} \alpha_i(z)\left(\sum_{m=1}^{\infty} w_m \lambda_m^{\frac{1}{2}}\psi_m(s_i') - \sum_{m=1}^{\infty} w_m \lambda_m^{\frac{1}{2}}\overline{\psi}_m(z_i)\right)$$

$$= \sum_{m=1}^{\infty} w_m \lambda_m^{\frac{1}{2}} \sum_{i=1}^{n} \alpha_i(z)\left(\psi_m(s_i') - \overline{\psi}_m(z_i)\right)$$

$$= \sum_{m=1}^{M} w_m \lambda_m^{\frac{1}{2}} \sum_{i=1}^{n} \alpha_i(z)\left(\psi_m(s_i') - \overline{\psi}_m(z_i)\right)$$

$$+ \sum_{m=M+1}^{\infty} w_m \lambda_m^{\frac{1}{2}} \sum_{i=1}^{n} \alpha_i(z)\left(\psi_m(s_i') - \overline{\psi}_m(z_i)\right).$$

We decomposed the noise-related error term into an infinite series corresponding to each eigenfunction $\psi_m$, $m = 1, 2, \cdots$, and partitioned that into the first $M$ elements and the rest. For each of the first $M$ elements, we can apply the standard confidence intervals for kernel ridge regression. Specifically, Corollary 1 in Whitehouse et al. (2024) implies that, with probability at least $1 - \delta/M$, we have

$$\sum_{i=1}^{n} \alpha_i(z)(\psi_m(s_i') - \overline{\psi}_m(z_i)) \leq \frac{\psi_{\max}\sigma_n(z)}{\sqrt{\rho}}\left(2\log\left(\sqrt{\frac{M}{\delta}\det(I + \rho^{-1}K_n)}\right)\right)^{\frac{1}{2}}.$$

Summing up over the first $M$ elements, and using a probability union bound, with probability at least $1 - \delta$, we have

$$\sum_{m=1}^{M} w_m \lambda_m^{\frac{1}{2}} \sum_{i=1}^{n} \alpha_i(z)(\psi_m(s_i') - \overline{\psi}_m(z_i))$$

$$\leq \sum_{m=1}^{M} w_m \lambda_m^{\frac{1}{2}} \frac{\psi_{\max}\sigma_n(z)}{\sqrt{\rho}}\left(2\log\left(\sqrt{\frac{M}{\delta}\det(I + \rho^{-1}K_n)}\right)\right)^{\frac{1}{2}}$$

$$\leq \frac{\psi_{\max}\sigma_n(z)}{\sqrt{\rho}}\left(\sum_{m=1}^{M} w_m^2\right)^{\frac{1}{2}}\left(\sum_{m=1}^{M} \lambda_m\right)^{\frac{1}{2}}\left(2\log\left(\sqrt{\frac{M}{\delta}\det(I + \rho^{-1}K_n)}\right)\right)^{\frac{1}{2}}$$

$$\leq \frac{C_v\psi_{\max}\sigma_n(z)}{\sqrt{\rho}}\left(\sum_{m=1}^{M} \lambda_m\right)^{\frac{1}{2}}\left(2\log\left(\sqrt{\frac{M}{\delta}\det(I + \rho^{-1}K_n)}\right)\right)^{\frac{1}{2}},$$

where the second inequality follows from the Cauchy-Schwarz inequality, and the third inequality follows from the bound $\|v\|_{\mathcal{H}_k} \leq C_v$.

For the rest of the elements $m > M$, we have

$$\sum_{m=M+1}^{\infty} w_m \lambda_m^{\frac{1}{2}} \sum_{i=1}^{n} \alpha_i(z)\left(\psi_m(s_i') - \overline{\psi}_m(z_i)\right) \leq 2\psi_{\max} \sum_{m=M+1}^{\infty} w_m \lambda_m^{\frac{1}{2}} \sum_{i=1}^{n} \alpha_i(z)$$

$$\leq 2\psi_{\max} \sum_{m=M+1}^{\infty} w_m \lambda_m^{\frac{1}{2}}\left(n\sum_{i=1}^{n} \alpha_i^2(z)\right)^{\frac{1}{2}}$$

$$\leq \frac{2\psi_{\max}\sigma_n(z)\sqrt{n}}{\sqrt{\rho}} \sum_{m=M+1}^{\infty} w_m \lambda_m^{\frac{1}{2}}$$

$$\leq \frac{2\psi_{\max}\sigma_n(z)\sqrt{n}}{\sqrt{\rho}}\left(\sum_{m=M+1}^{\infty} w_m^2\right)^{\frac{1}{2}}\left(\sum_{m=M+1}^{\infty} \lambda_m\right)^{\frac{1}{2}}$$

$$\leq \frac{2C_v\psi_{\max}\sigma_n(z)}{\sqrt{\rho}}\left(n\sum_{m=M+1}^{\infty} \lambda_m\right)^{\frac{1}{2}}.$$

The first inequality holds by the definition of $\psi_{\max}$. The second inequality follows from the Cauchy-Schwarz inequality. The third inequality is derived using Lemma 1. The fourth inequality again applies the Cauchy-Schwarz inequality, and the final inequality results from the upper bound on the RKHS norm of $v$.

Putting all the terms together, with probability $1 - \delta$,

$$|f(z) - \hat{f}_n(z)| \leq \beta(\delta)\sigma_n(z),$$

where $\beta(\delta) =$

$$C_f + \frac{C_v \psi_{\max}}{\sqrt{\rho}} \left( \sum_{m=1}^{M} \lambda_m \right)^{\frac{1}{2}} \left( 2 \log \left( \sqrt{\frac{M}{\delta}} \det(I + \rho^{-1} K_n) \right) \right)^{\frac{1}{2}} + \frac{2C_v \psi_{\max}}{\sqrt{\rho}} \left( n \sum_{m=M+1}^{\infty} \lambda_m \right)^{\frac{1}{2}},$$

that completes the proof.

# 7 Proof of Theorem 3

To analyze the performance of KUCB-RL, we first define an event $\mathcal{E}$ that all the confidence intervals used in the algorithm hold true.

$$\mathcal{E} = \left\{ |f_t(z) - \hat{f}_t(z)| \leq \beta(\delta)\sigma_t(z), \quad \forall t \in [T] \right\}, \tag{13}$$

where

$$\beta(\delta) = \mathcal{O} \left( w + \frac{w}{\sqrt{\rho}} \sqrt{\log \left( \frac{T}{\delta} \right) + \gamma(T, \rho)} \right).$$

By Theorem 1, we have $\Pr[\mathcal{E}] \geq 1 - \delta/2$. We note the under Assumption 4, $\|v\|_{\mathcal{H}_{k'}} \leq C_v = \mathcal{O}(w)$. Also, for $v : \mathcal{S} \to [0, w]$, we have $\|Pv\|_{\mathcal{H}_k} = \mathcal{O}(w)$. See Yeh et al. (2023), Lemma 3, for a proof. Since $v_t$ is upper bounded by $w$ by construction, we have $\|Pv_t\| = \mathcal{O}(w)$ that replaces $C_f$ in the expression of $\beta$ in Theorem 1.

We condition the rest of the proof on event $\mathcal{E}$.

Consider $t_0$ such that $(t_0 \mod w) = 0$ we bound the regret over window $t \in [t_0 + 1, t_0 + w]$, denoted by $\mathcal{R}_{t_0}(w)$. In addition let $V_w^\star(s)$ denote the optimum achievable total reward over a window of size $w$ starting with initial state $s$, and $V_w^\pi(s)$ denote the total reward over a window of size $w$ achieved by KUCB-RL starting with initial state $s$.

$$\mathcal{R}_{t_0}(w) = wJ^\star - \sum_{t=t_0+1}^{t_0+w} r(s_t, a_t) = wJ^\star - V_w^\star(s_{t_0+1}) + V_w^\star(s_{t_0+1}) - \sum_{t=t_0+1}^{t_0+w} r(s_t, a_t).$$

For a bounded function $v : \mathcal{S} \to \mathbb{R}$, we define its span as $\text{span}(v) = \sup_{s,s' \in \mathcal{S}} |v(s) - v(s')|$.

The first term is bounded by the span of $v^*$.

**Lemma 2.** *For any $s$, $|wJ^\star - V_w^\star(s)| \leq span(v^*)$.*

Proof follows the exact same lines as in the proof of Lemma 13 in Wei et al. (2021).

We next bound the second term in $\mathcal{R}_{t_0}(w)$. We first prove that $V_w^\star(s) \leq v_{t_0}(s)$.

**Lemma 3.** *Under event $\mathcal{E}$, we have $V_w^\star(s) \leq v_{t_0}(s)$, $\forall s \in \mathcal{S}$.*

**Proof.** We can prove this by induction. Note that $V_0^\star(s) = v_{t_0+w+1}(s) = 0$. For any $j \in [w]$, we have

$$
\begin{aligned}
V_j^\star(s) - v_{t_0+w+1-j}(s) &= \max_{a \in \mathcal{A}} Q_j^\star(s, a) - \max_{a' \in \mathcal{A}} q_{t_0+w+1-j}(s, a') \\
&\leq \max_{a \in \mathcal{A}} \{ Q_j^\star(s, a) - q_{t_0+w+1-j}(s, a) \} \\
&= \max_{a \in \mathcal{A}} \{ [PV_{j+1}^\star](s, a) - [Pv_{t_0+w-j}](s, a) \} \\
&= \max_{a \in \mathcal{A}} \{ \mathbb{E}_{s' \sim P(\cdot|s,a)} [V_{j+1}^\star(s') - v_{t_0+w-j}(s')] \} \\
&\leq 0.
\end{aligned}
$$

The first inequality is due to rearrangement of $\max$, and the second inequality is by the induction assumption. We thus have $V_w^\star(s) \leq v_{t_0}(s)$.

$\square$

We now bound the difference between $v_{t_0}(s_{t_0+1})$ and sum of the reward over the window starting at step $t_0 + 1$: $v_{t_0+1}(s_{t_0+1}) - V_w^\pi(s_{t_0+1})$. We note that $v_{t_0+w}(s_{t_0+w}) = V_0^\pi(s_{t_0+w}) = 0$ and

$$
\begin{aligned}
v_{t_0+j}(s_{t_0+j}) - V_{w-j}^\pi(s_{t_0+j}) &= q_{t_0+j}(s_{t_0+j}, a_{t_0+j}) - Q_{w-j}^\pi(s_{t_0+j}, a_{t_0+j}) \\
&\leq [Pv_{t_0+j+1}](s_{t_0+j}, a_{t_0+j}) - [PV_{w-j}^\pi](s_{t_0+j}, a_{t_0+j}) + 2\beta(\delta)\sigma_{t_0}(s_{t_0+j}, a_{t_0+j}) \\
&= v_{t_0+j+1}(s_{t_0+j+1}) - V_{w-j-1}^\pi(s_{t_0+j+1}) + 2\beta(\delta)\sigma_{t_0}(s_{t_0+j}, a_{t_0+j}) \\
&\quad + ([Pv_{t_0+j+1}](s_{t_0+j}, a_{t_0+j}) - v_{t_0+j+1}(s_{t_0+j+1})) \\
&\quad + \left( V_{w-j-1}^\pi(s_{t_0+j+1}) - [PV_{w-j}^\pi](s_{t_0+j}, a_{t_0+j}) \right).
\end{aligned}
$$

The inequality holds under event $\mathcal{E}$. We obtained a recursive relationship for $v_{t_0+j}(s_{t_0+j}) - V_{w-j}^\pi(s_{t_0+j})$. Iterating over $j = w$ to $j = 1$, we get

$$
\begin{aligned}
v_{t_0+1}(s_{t_0+1}) - V_w^\pi(s_{t_0+1}) &\leq \sum_{t=t_0+1}^{t_0+w} 2\beta(\delta)\sigma_{t_0}(s_t, a_t) + \sum_{t=t_0+1}^{t_0+w} ([Pv_{t+1}](s_t, a_t) - v_{t+1}(s_{t+1})) \\
&\quad + \sum_{t=t_0+1}^{t_0+w} \left( V_{w+t_0-t-1}^\pi(s_{t+1}) - [PV_{w+t_0-t}^\pi](s_t, a_t) \right).
\end{aligned}
$$

The second and third terms are zero mean martingales with a span of $2w$, which are sub-Gaussian random variables with parameter $w$. Therefore, by Azuma-Hoeffding inequality (Lalley, 2013), with probability at least $1 - \delta/2$,

$$
\begin{aligned}
\sum_{t=1}^{T} ([Pv_{t+1}](s_t, a_t) - v_{t+1}(s_{t+1})) &+ \sum_{t=1}^{T} \left( V_{w+w\lfloor(t-1)/w\rfloor-t-1}^\pi(s_{t+1}) - [PV_{w+w\lfloor(t-1)/w\rfloor-t}^\pi](s_t, a_t) \right) \\
&\leq w\sqrt{2T\log\left(\frac{2}{\delta}\right)}.
\end{aligned}
$$

We note that for each $t \in [T]$, we can present the corresponding $t_0$ with $t_0 = w\lfloor(t-1)/w\rfloor$. Summing up the regret over all windows of size $w$ up to time $t$, we have, with probability $1 - \delta$,

$$
\mathcal{R}(T) \leq \frac{T\,\text{span}(v^*)}{w} + w\sqrt{2T\log\left(\frac{2}{\delta}\right)} + 2\beta(\delta) \sum_{t=1}^{T} \sigma_{w\lfloor(t-1)/w\rfloor}(z_t). \tag{14}
$$

It thus remains to bound $\sum_{t=1}^{T} \sigma_{w\lfloor(t-1)/w\rfloor}(z_t)$.

The sum of sequential standard deviations of a kernel based model is often bounded using the following result from Srinivas et al. (2010) that is similar to the elliptical potential lemma in linear bandits (see, Abbasi-Yadkori et al., 2011).

$$\sum_{t=1}^{T} \sigma_{t-1}^2(z_t) \leq \frac{2\gamma(T;\rho)}{\log(1+1/\rho)}. \tag{15}$$

This result however is not directly applicable here due to the $w\lfloor(t-1)/w\rfloor$ subscript in $\sigma_{w\lfloor(t-1)/w\rfloor}$ rather $\sigma_{t-1}$. A loose approach would be to partition the sequence into $w$ sequences, each for one $j \in [w]$ of the form $\sigma_{w(i-1)+j}$, $i = 1, 2, \cdots T/w$. For each of those sequences, (15) is applicable and we get

$$\sum_{i=1}^{T/w} \sigma_{w(i-1)+j}^2(z_{wi+j}) \leq \frac{2\gamma(T/w;\rho)}{\log(1+1/\rho)}. \tag{16}$$

Using this bound we have

$$\sum_{t=1}^{T} \sigma_{w\lfloor(t-1)/w\rfloor}^2(z_t) = \sum_{j=1}^{w} \sum_{i=1}^{T/w} \sigma_{w(i-1)+j}^2(z_{wi+j})$$
$$\leq \frac{2w\gamma(T/w;\rho)}{\log(1+1/\rho)}. \tag{17}$$

Next, we prove a stronger bound on $\sum_{t=1}^{T} \sigma_{w\lfloor(t-1)/w\rfloor}(z_t)$ that contributes to the sublinear regret bounds in this paper.

**Lemma 4.** *For a sequence of observation points $\{z_t\}_{t=1}^{T}$ and any $w \in \mathbb{N}$, we have*

$$\sum_{t=1}^{T} \sigma_{w\lfloor(t-1)/w\rfloor}(s_t, a_t) \leq \sqrt{\frac{2\gamma(T;\rho)}{\log(1+1/\rho)}\left(T + \frac{2w^2\gamma(T/w;\rho)}{\log(1+1/\rho)}\right)}. \tag{18}$$

*Proof of Lemma 4.* We use the following lemma on the ratio of variances conditioned on two sets of observations.

**Lemma 5** (Proposition A.1 in Calandriello et al. (2022)). *For any sequence of points $\{z_j\}_{j=1}^{T}$, for any $z$ and $t' < t$*

$$1 \leq \frac{\sigma_{t'}^2(z)}{\sigma_t^2(z)} \leq 1 + \sum_{j=t'+1}^{t} \sigma_{t'}^2(z_j).$$

We thus can write

$$\sum_{t=1}^{T} \sigma_{w\lfloor(t-1)/w\rfloor}(s_t, a_t) \leq \sum_{t=1}^{T} \sigma_t(s_t, a_t)\sqrt{1 + \sum_{j=w\lfloor(t-1)/w\rfloor+1}^{t} \sigma_{w\lfloor(t-1)/w\rfloor}^2(s_j, a_j)}$$
$$\leq \sqrt{\sum_{t=1}^{T} \sigma_t^2(s_t, a_t)}\sqrt{T + w\sum_{t=1}^{T} \sigma_{w\lfloor(t-1)/w\rfloor}^2(s_t, a_t)}$$
$$\leq \sqrt{\frac{2\gamma(T;\rho)}{\log(1+1/\rho)}\left(T + \frac{2w^2\gamma(T/w;\rho)}{\log(1+1/\rho)}\right)}. \tag{19}$$

The first inequality is by Lemma 5, the second inequality follows from Cauchy-Schwarz inequality, and the last inequality is the bound established in Equation (17). This completes the proof of Lemma 4.

$$\square$$

Using Lemma 4, and substituting the value of $\beta(\delta)$ into (14), we obtain

$$\mathcal{R}(T) = \mathcal{O}\left(\frac{T}{w} + \left(w + \frac{w}{\sqrt{\rho}}\sqrt{\gamma(T;\rho) + \log\left(\frac{T}{\delta}\right)}\right)\sqrt{\rho T\gamma(T;\rho) + \rho^2 w^2\gamma(T;\rho)\gamma(T/w;\rho)}\right).$$

(20)

The proof of the regret bound is complete.

## 8 Mercer Theorem and the RKHSs

Mercer theorem (Mercer, 1909) provides a representation of the kernel in terms of an infinite dimensional feature map (e.g., see, Christmann and Steinwart, 2008, Theorem 4.49). Let $\mathcal{Z}$ be a compact metric space and $\mu$ be a finite Borel measure on $\mathcal{Z}$ (we consider Lebesgue measure in a Euclidean space). Let $L^2_\mu(\mathcal{Z})$ be the set of square-integrable functions on $\mathcal{Z}$ with respect to $\mu$. We further say a kernel is square-integrable if

$$\int_{\mathcal{Z}}\int_{\mathcal{Z}} k^2(z, z')\, d\mu(z)d\mu(z') < \infty.$$

**Theorem 5** (Mercer Theorem). *Let $\mathcal{Z}$ be a compact metric space and $\mu$ be a finite Borel measure on $\mathcal{Z}$. Let $k$ be a continuous and square-integrable kernel, inducing an integral operator $T_k :$ $L^2_\mu(\mathcal{Z}) \to L^2_\mu(\mathcal{Z})$ defined by*

$$(T_k f)(\cdot) = \int_{\mathcal{Z}} k(\cdot, z')f(z')\, d\mu(z'),$$

*where $f \in L^2_\mu(\mathcal{Z})$. Then, there exists a sequence of eigenvalue-eigenfeature pairs $\{(\lambda_m, \varphi_m)\}_{m=1}^{\infty}$ such that $\lambda_m > 0$, and $T_k\varphi_m = \lambda_m\varphi_m$, for $m \geq 1$. Moreover, the kernel function can be represented as*

$$k(z, z') = \sum_{m=1}^{\infty} \lambda_m\varphi_m(z)\varphi_m(z'),$$

*where the convergence of the series holds uniformly on $\mathcal{Z} \times \mathcal{Z}$.*

According to the Mercer representation theorem (e.g., see, Christmann and Steinwart, 2008, Theorem 4.51), the RKHS induced by $k$ can consequently be represented in terms of $\{(\lambda_m, \varphi_m)\}_{m=1}^{\infty}$.

**Theorem 6** (Mercer Representation Theorem). *Let $\{(\lambda_m, \varphi_m)\}_{i=1}^{\infty}$ be the Mercer eigenvalue eigenfeature pairs. Then, the RKHS of $k$ is given by*

$$\mathcal{H}_k = \left\{f(\cdot) = \sum_{m=1}^{\infty} w_m\lambda_m^{\frac{1}{2}}\varphi_m(\cdot) : w_m \in \mathbb{R}, \|f\|^2_{\mathcal{H}_k} := \sum_{m=1}^{\infty} w_m^2 < \infty\right\}.$$

Mercer representation theorem indicates that the scaled eigenfeatures $\{\sqrt{\lambda_m}\varphi_m\}_{m=1}^{\infty}$ form an orthonormal basis for $\mathcal{H}_k$.

