# OpenReview forum: "Kernel-Based Function Approximation for Average Reward Reinforcement Learning: An Optimist No-Regret Algorithm"
_NeurIPS.cc/2024/Conference — NeurIPS 2024 poster_

### Official Review · Reviewer_unzJ · 2024-07-12

**Soundness:** 2
**Presentation:** 3
**Contribution:** 3
**Rating:** 6
**Confidence:** 2

**Summary:**

This paper studies learning in MDPs with the long-run average reward objective, assuming that the MDP satisfies some kernel assumptions. A UCB-type of learning algorithm is proposed and is proved to achieve sublinear regret when the eigenvalues of the kernel operators decay at the rate of a p-th degree polynomial with p > 1.
The paper also proves a confidence interval bound for kernel ridge regression, which is of independent interest.

**Strengths:**

The result in this paper seems to be strong: it proves a regret bound with a pretty weak MDP assumption (weaker than weakly-communicating), and a RKHS kernel assumption on the structure of the transition kernel which seems to significantly weaken the tabular and linear assumptions in prior work.

**Weaknesses:**

The use of Bachmann-Landau notation is confusing. Specifically, in Assumption 3, it is unclear which variable is scaling in the o(1) notation. Similarly, in Theorem 2, multiple variables appear in the big-O notation, again making it unclear which variable is scaling.
It seems to me that the use of big-O notation is unnecessary, given that most steps in the proofs are non-asymptotic. It would be clearer to state the theorems in the non-asymptotic form, and then state a corollary when certain variables scale.

Details for bounding the maximal information gain \gamma(\rho; t) is missing. The authors mention that this maximum information gain is O(d \log(t)) in line 216-211 in the special case of d-dim linear kernels, but did not provide references or proofs.

**Questions:**

M is not defined in the equation in line 291. Do you mean the equation holds for any natural number M?

**Limitations:**

A discussion of limitation is included.

---

> ### Author Rebuttal · Authors · 2024-08-06
>
> Thank you for reviewing our paper. We appreciate your positive feedback on the strength of the results. Here, we address your comments in detail and hope this will enhance your evaluation of the paper.
>
> > *The use of $O$ notation:*
>
> As mentioned by the reviewer all of our proofs are non-asymptotic and our use of $O$ notation, which is common practice in the literature, serves to simplify presentations. Theorem 1 does not use $O$ notation. We will revise the statement of Assumption 3 to read: "...$M\sum_{m=M+1}^{\infty}\lambda_m\le C$, for any $M\in\mathbb{N}$ and some constant $C>0$ independet of $M$." We will also clarify the statement of Theorem 2 by introducing an absolute constant independent of all parameters: $T,w, \rho, \delta$. We are ultimately interested in scaling with $T$ after $w$ and $\rho$ are selected optimally with respect to $T$, as given in Remark 2, which we will further clarify.
>
> > *References for bounds on information gain*
>
> Thank you for pointing out the missing references for maximum information gain. The bound for the linear case is given in [22]. Further discussions and tight bounds for kernels with exponential and polynomial eigenvalue decay are provided in [35]. We will ensure these references are properly included.
>
> > *M is not defined in the equation in line 291. Do you mean the equation holds for any natural number M?*
>
> Yes, the statement holds for any natural number $M$. We will clarify this point in the revision.
>
> For further details on the role of this $M$ in the analysis, please see our response to Reviewer KBUd. Additionally, a simplified presentation of the confidence interval is given in Remark 1, where $M$ is removed.
>
> We would be happy to provide further clarifications on any of these points.

---

> > ### Comment · Reviewer_unzJ · 2024-08-12
> >
> > Thanks for your response. I have one more question: I am a bit confused if "$M\sum_{m=M+1}^\infty \lambda_m \leq C$ for any $M$" is the definition of Assumption 3, for two reasons:
> >
> > First, $o(1)$ typically means a sequence converges to zero, while the definition you just gave only requires the sequence to be bounded. Did you intend to write $O(1)$ instead of $o(1)$ in Assumption 3?
> >
> > Second, in the sentence after Assumption 3, you said that Assumption 3 is implied by p-polynomial eigendelay with $p> 1$, which I do not quite follow.
> > By definition 1 of [35], p-polynomial eigendelay means $\lambda_m \leq C'*m^{-p}$ for some constant $C'$, so $M\sum_{m=M+1}^\infty \lambda_m$ has order $\Theta(M^{-p+2})$, which is bounded only when $p\geq 2$.

---

> > > ### Author Response · Authors · 2024-08-12
> > >
> > > That is correct, and thank you for pointing it out. We overlooked this rather straightforward calculation. Here, we reexamine the entire calculation for clarity.
> > >
> > > For Remark 1, we select $M$ with respect to $n$ such that the third term in $\beta(\delta)$ becomes only a constant. We have updated Assumption 3 as follows:
> > >
> > > **Assumption 3** For kernel k′, we assume $\sum_{m=M+1}^{\infty}\lambda_m \le C_1M^{-\alpha}$ for constants $C_1, \alpha>0$, both independnet of $M$.
> > >
> > > This is a mild assumption. For instance, for the $p$-polynomial kernels defined in Definition 1, which apply to a large class of common kernels including SE, Matérn, and NT kernels, Assumption 3 is satisfied with $C_1=C/(p-1)$ and $\alpha=p-1$, where $C>0$ is the constant in Definition 1.
> > >
> > > Under this adjustment, to prove Remark 1, we choose $M=\lceil n^{1/\alpha}\rceil$. In the third term in $\beta(\delta)$, we have $n\sum_{m=M+1}^{\infty}\lambda_m\le C_1nM^{-\alpha}\le C_1$, thus the third term is only a constant. In the second term in $\beta(\delta)$ rather than $\log(\frac{n}{\delta})$ under square root, we have $\frac{1}{\alpha}\log(n)+\log(\frac{1}{\delta})$, where $\alpha$ is only a kernel specific constant independent of all other paramters $n, C_f, \delta, w, \rho$.
> > >
> > > We appreciate your feedback and would be grateful if you could please let us know if this adjustment and explanation address your question.

---

> ### Comment · Reviewer_unzJ · 2024-08-12
>
> Thanks. I am convinced that this new version of Assumption 3 is implied by p-eigendecay with $p>1$, and the expression of $\beta(\delta)$ in remark 1 remains unchanged. However, since Theorem 2 also relies on Assumption 3, I am a bit concerned if this change in Assumption 3 affects the correctness of Theorem 2. Could you please clarify to what extent Theorem 2 relies on Assumption 3? In particular, is there a more general version of Theorem 2 that does not rely on Assumption 3 and has $\beta(\delta)$ in its regret bound?
>
> Also, I notice that on line 527, the expression of beta used in the proof of Theorem 2 does not exactly match the expression of beta in remark 1 (one starts with $C_f$ and the other starts with $w$). Could you please clarify?

---

> > ### Author Response · Authors · 2024-08-13
> >
> > Thank you for your response.
> >
> > Theorem 2 relies on Assumption 3 only through Remark 1. As discussed above, the adjustment to Assumption 3 does not affect the presentation of $\beta(\delta)$ in Remark 1, nor does it affect Theorem 2.
> >
> > In Theorem 2, the first factor in the second term inside the $O$ notation represents $\beta(\delta)$. Specifically, $w + \frac{w}{\sqrt{\rho}} \sqrt{\gamma(\rho; T) + \log(T/\delta)}$ can be replaced by $\beta(\delta)$, offering a variation of the theorem with $\beta(\delta)$. This term matches the expression of $\beta(\delta)$ in Line 527, as pointed out by the reviewer.
> >
> > The parameter $C_f$ in Theorem 1 denotes the upper bound on the RKHS norm of the target function $f$. Within the proof of Theorem 2, the target function is $Pv_t$. We substitute $C_f$ with $w$ because $||Pv_t||_{H_k} = O(w)$, given that $v_t$ is bounded by $w$ (due to the projection step onto $[0,w]$), and according to Lemma 3 of [42], that:
> >
> > $||Pv||_{H_k} = O(span(v))$.
> >
> > This is explained in Lines 530 and 531. Following your comment, we will add more details and reiterate this point in the main text.
> >
> > Thank you for your engagement in the review process, which has significantly contributed to improving the paper. We appreciate your feedback and would be grateful if you could let us know if our response has clarified your question.

---

> > > ### Comment · Reviewer_unzJ · 2024-08-13
> > >
> > > I appreciate the authors' thorough response to my concerns. Based on the rebuttal, I will raise my score to 6.

---

> > > > ### Author Response · Authors · 2024-08-14
> > > >
> > > > Thank you again for your thoughtful comments. We appreciate your contribution to the review process.

---

### Official Review · Reviewer_pN55 · 2024-07-16

**Soundness:** 2
**Presentation:** 2
**Contribution:** 3
**Rating:** 5
**Confidence:** 2

**Summary:**

The authors consider the setting of reinforcement learning in infinite horizon average reward MDPs. In particular, they consider a kernel based function approximation to represent value functions. Most of the prior work involving kernel regression has been in the context of bandits, where the state cardinality is 1. In the context of reinforcement learning, the complexity is enhanced, since the state space is no longer degenerate and actions influence transitions to future states. Hence, the setting in which kernel approximation is employed is thus more complicated. Some limited prior work in the domain of RL involves using kernel functions whose eigenvalues decay in an exponential manner which in the current work has been relaxed to polynomial decay, thus encapsulating a wider class of approximation architectures. The value functions are computed using kernel functions along with a confidence bound. Another contribution of the authors is characterizing the confidence bounds in this context. The value functions are set to zero ahead of a time horizon $w$ and are recursively back calculated using samples generated till previous time horizon. The next set of samples are generated using these new estimates of value functions. The final regret bounds are characterized as a function of the polynomial decay $p$, but are sublinear for all $p>1$.

**Strengths:**

1. A major strength of using kernel functions is the expressibility of the function class with respect of the representation of state action and state value functions. Most literature in the context of RL relies on linear function approximation (which is a special case of kernel functions) due to ease of analysis. Kernel approximations on the other hand capture functions classes such as neural tangent kernel and Matern family. Moreover this work relaxes the exponential decay of eigenvalues utilized in the previous work to polynomial decay, which is a significant improvement.

2. The final bounds are in general form which depends on the polynomial decay of the eigenvalues. The authors also remark on how the regret bounds for the linear function approximation case can be derived from this general form. The authors also present the first sublinear regret bounds in the context of polynomially decaying eigenvalues of the kernel function.

3. Their construction of confidence bounds may be of independent interest for the design of optimistic algorithms in RL using kernel function approximation.

**Weaknesses:**

1. The function approximation representation of the state action value function at time $t$ involves inversion of a matrix of dimension $t$. This operation is performed at every iteration of the algorithm. As $t$ grows, this computation might be expensive. It is unclear as to whether there is a short cut to circumvent this step in the algorithm, which might make it intractable in large time horizons.

2. The value functions are being reset to 0 repeatedly after a finite time horizon $w$. The samples generated till that time instant only manifest in the confidence bounds. It's not entirely clear as to what the significance of reseting this value function to 0 is establishing since it seems to be not data-efficient.

**Questions:**

1. The notion of maximum information gain $\gamma$ is unclear. If the information gained is maximized, intuitively, it results in better regret bounds? However, larger the value of $\gamma$, greater the magnitude of the regret. It would help to characterize what $\gamma$ represents in the context of RL.

2. It is unclear as to how the reseting and back calculation of the value function from time $t+w+1$ to $t+1$ in Algorithm 1 helps with the analysis.

2. Won't the projection operator in Equation 5 negate the role of the confidence bounds? It makes sense for the value functions to not exceed the horizon $w$, since the rewards are $<1$, however the information from the previous samples seem to be represented solely through the confidence bound term $\beta \sigma$. Would the performance be better if the projection was onto a space larger than $w$? Since there seems to be a tradeoff with respect to $w$ in the final bounds, is there a intuitive explanation as to why such a tradeoff exists? As in what aspect of the analysis gets better as $w$ grows larger and what gets worse?

**Limitations:**

The limitations of this work have been explicitly addressed.

---

> ### Author Rebuttal · Authors · 2024-08-06
>
> Thank you for your detailed review of our paper. We appreciate your positive feedback regarding the generality of our setting and results, as well as the potential independent interest in the confidence bounds. We will address your comments and questions in detail, hoping this will enhance your evaluation of the paper.
>
> > *Computational complexity:*
>
> Kernel-based models provide powerful and verstile nonlinear function approximations and lend themselves to theoretical analysis.
> It is, however, well-known that kernel ridge regression has a relatively high computational complexity, primarily due the matrix inversion step. This challenge is not unique to our work and is common across kernel-based supervised learning, bandit, and RL literature. Luckily, sparse approximation methods such as Sparse Variational Gaussian Processes (SVGP) or the Nyström method significantly reduce the computational complexity (to as low as linear in some cases), while maintaining the kernel-based confidence intervals and, consequently, the eventual rates; e.g., see Vakili et al. ICML'22 and references therein. These results are, however, generally applicable and not specific to our problem. We thus preferred to avoid further complicating the notation and presentation of an already notation-heavy RL setting to focus on our main results and contributions specific to this paper.
>
> ``Improved Convergence Rates for Sparse Approximation Methods in Kernel-Based Learning'', Vakili, Scarlett, Shiu, Bernacchia, ICML 2022.
>
> Additionally, as a minor point, the matrix inversion step is required only every other $w$ steps, in contrast to every step, which further improves computational complexity. We will add these discussions to the revision.
>
> > 1. *maximum information gain:*
>
> This is a kernel-specific quantity that captures the complexity of kernel-based learning. It is a common measure in the literature on kernel-based bandits [22, 23, 24, 36, 27] and RL [12, 25, 26]. As defined in Section 2.5, the value of $\gamma$ depends only on the kernel and the free parameter $\rho$. The algorithm itself does not aim to maximize information gain; rather, it is an algorithm-independent and kernel-specific complexity term that appears in the regret bound. Intuitively, $\gamma$ captures the complexity of the function class. Under a linear assumption, $\gamma=O(d\log(T))$. For kernels with exponentially decaying eigenvalues  $\gamma=O(poly\log(T))$, and for kernels with polynomially decaying eigenvalues, such as the Matérn family and Neural Tangent kernels, the bound on $\gamma$ is given in Section 2.5. For proofs of these bounds, see [22, 35].
>
> > 2. *Reseting and back calculation of the value function on a window with size $w$:*
>
> We here address this question along with the related observations from the "Weaknesses" section and the subsequent discussion on the tradeoff in choosing $w$ from the next question. We understand the reviewer's comments to be highlighting two specific aspects of our algorithm and analysis.
>
> **First aspect:** For kernel ridge regression on $[P v_{t+1}]$, we only use observations up to $t_0$, where $(t_0 \mod w) = 0$ and $t_0+1 \le t \le t_0+w$. This might seem data inefficient as it does not utilize the $t-t_0 \le w$ samples:
>
> We address this in our analysis. After Theorem 2, which presents the performance of the algorithm, it is noted: "Algorithm 1 updates the observation set every $w$ steps, requiring us to characterize and bound the effect of this delay in the proof. A straightforward application of the elliptical potential lemma results in loose bounds that do not guarantee no-regret. We establish a tighter bound on this term, contributing to the improved regret bounds." In particular, we show that this delay in updating the observation set does not change the eventual rates. The proof, based on Lemma 4, is detailed in lines 555 - 565 in the appendix. The key idea in the proof is to bound the ratio between a delayed uncertainty estimate update and a non-delayed one, followed by some arithmetic manipulation.
>
> **Second aspect:** Regarding the window’s role (i.e., setting the value function to $0$ at the end of the window, backtracking to compute the proxy value function over the window, and unrolling the policy in a forward direction over the window), there is an apparent tradeoff in choosing the window size:
>
> This tradeoff balances between the strength of the value function and the strength of the noise. A longer window is preferred to capture the long-term performance of the policy, while a larger window increases the observation noise affecting the kernel ridge regression. We recall that both target function $[Pv_{t+1}]$ and observation noise $v_t-[Pv_{t+1}]$ are bounded by $w$. This tradeoff is explicitly seen in the regret bounds. The optimal size of the window, as specified in line 335, results from an optimal interplay between these two factors, which is explicitly captured in the regret bounds.
>
> We appreciate this discussion and believe that the inclusion of these points will enhance the paper. Please let us know if any further clarifications are needed regarding these aspects.

---

> ### Author Response · Authors · 2024-08-06
> **Rebuttal by Authors - Part 2**
>
> > 3. *The projection operator in Equation 5:*
>
> The analysis requires a confidence interval with diminishing width. In other words, we condition the proof on the validity of the confidence intervals with probability $1-\delta$, hence "... with probability at least $1 − \delta$ ..." in the statement of Theorem 2. We also show that the regret grows with the sum of the widths of confidence intervals over time.
>
> Projecting on $[0,w]$ maintains the validity of confidence intervals. It also does not affect the growth rate of the sum of the widths: $\sum_{t=1}^T\sigma_{\lfloor t/w\rfloor}(z_t)$ given that the uncertainties eventually diminish in a way that the confidence interval width $\beta(\delta)\sigma_{\lfloor t/w\rfloor}(z_t)$ becomes smaller than $w$. However, it significantly simplifies the proof by allowing us to use a uniform $w$ upper bound on noise over time, rather than dealing with noise terms of the power $\beta(\delta)\sigma_{\lfloor t/w\rfloor}(z_t)$ that could be larger than $w$ for small $t$.
>
> In summary, although removing projections does not seem to affect the eventual rates, it can complicate the proof. We also note that the projection of the confidence intervals onto a priori known interval for the target function is a commonly used technique across RL literature [9, 10, 12, 23, 25, 29] and is not specific to our work.
>
> We appreciate your detailed comments and constructive review. We would be happy to provide any further clarifications or engage in further discussions.

---

> > ### Comment · Reviewer_pN55 · 2024-08-12
> >
> > Thank you for your response to my comments. I have read them and am keeping my score.

---

### Official Review · Reviewer_KBUd · 2024-07-18

**Soundness:** 3
**Presentation:** 3
**Contribution:** 3
**Rating:** 7
**Confidence:** 2

**Summary:**

The paper proposes a kernel-based optimistic algorithm for the average reward setting and corresponding regret analysis. As described, the kernel-based setting is a more general extension of linear structure to an infinite-dimensional linear model in the feature space of a positive definite kernel.

**Strengths:**

This seems to be the first treatment for average reward MDP in the kernel-based setting. The paper seems to be well-written. Overall, this could be interesting addition to the set of literature in average reward RL

**Weaknesses:**

The technical portion of the main text can be improved if the paper discusses the key steps/challenges of the proof. E.g., Theorem 1 is based on results of prior work [37]; without moving to the appendix, it is impossible to extract the technical novelty in this paper.

**Questions:**

1)Can the results be improved to \sqrt{T} under uniform mixing conditions (which is a stronger assumption)?

---

> ### Author Rebuttal · Authors · 2024-08-06
>
> Thank you for reviewing our paper. We appreciate your positive feedback.
>
> Following your comment, we will enhance the main text with more detailed technical content. We have provided a proof sketch in the paragraph following Remark 1, which we will expand to further detail the proof of Theorem 1.
>
> Unlike typical settings such as (supervised) kernel ridge regression or kernel-based bandit settings [22, 23, 24, 36, 37, 24], in the RL setting, confidence intervals are applied to $f_t = [P v_{t+1}]$, which varies due to the Markovian nature of temporal dynamics, with each $v_t$ derived recursively from $v_{t+1}$. Hence, the confidence interval must be applicable to all functions $v$ in a function class. This requirement is captured in the theorem: "For all $z \in Z$ and $v: ||v||_{H_k'} \le C_v$, the following holds ...". To achieve such confidence intervals, we use a novel technique by leveraging the Mercer representation of $v$ and decomposing the prediction error $|f(z) - \hat{f}_n(z)|$ into error terms corresponding to each Mercer eigenfunction $\psi_m$. We then partition these terms into the first $M$ elements corresponding to eigenfunctions with the largest $M$ eigenvalues and the rest. For each of these $M$ eigenfunctions, we obtain high probability bounds using existing confidence intervals from [24]. Summing up over $m$, and using a bound based on uncertainty estimates, we achieve the high probability bound—corresponding to the second term in $\beta(\delta)$, and bound the remaining elements based on the eigendecay—corresponding to the third term in $\beta(\delta)$.
>
> In summary, the key technical novelties in deriving the concentration bound involve leveraging the Mercer decomposition of $v$ and the error term, partitioning the decomposition into the first $M$ elements and the rest, bounding each of the first $M$ elements using exisitng kernel-based confidence intervals, bounding the rest based on eigendecay, and then summing up over all $m$ to derive the confidence interval.
>
> We emphasize that this is a substantial result that can be applied across various RL problems.
>
> > *1) Can the results be improved to \sqrt{T} under uniform mixing conditions (which is a stronger assumption)?*
>
>
> As noted in the introduction, [20] achieved a regret bound of $\tilde{O}(\frac{1}{\sigma}\sqrt{t_{mix}^3 T})$ under the linear bias function assumption, where $\sigma$ is the smallest eigenvalue of the policy-weighted covariance matrix. While assuming a strictly positive smallest eigenvalue (independnt of $T$) for the covariance matrix is reasonable in a finite-dimensional setting where $d \ll T$, it becomes unrealistic in the kernel setting. This presents significant challenges in adapting the existing results to the kernel setting. It is not clear whether tighter results can be achieved under uniform mixing conditions, necessitating further research.
>
> Please let us know if any further clarifications on these points are required.

---

> ### Comment · Reviewer_KBUd · 2024-08-11
>
> Thanks for your detailed response. I am broadly satisfied by the rebuttal.

---

### Decision · Program_Chairs · 2024-09-25

**Decision:**

Accept (poster)

**Comment:**

This paper proposes a kernel-based optimistic algorithm for average reward RL in the infinite horizon setting. The authors provide new no-regret performance guarantees under kernel-based modeling assumptions, extending previous work to the more complex RL context.
The reviewers highlight both strengths and weaknesses of the paper. They praise its technical soundness and contribution, recognize its novelty in the average reward MDP context, and emphasize the importance of the kernel-based approach. However, some concerns are raised regarding computational complexity, the effectiveness of the value function reset, and the role of the projection operator in the analysis. The authors address all these issues.

Overall, the paper presents a solid contribution to the field of average reward RL by extending kernel-based methods and achieving sublinear regret bounds. The reviews acknowledge the novelty and theoretical strength of the work while raising valid concerns about computational complexity and specific aspects of the analysis. The authors demonstrate a comprehensive understanding of the issues and provide convincing responses. This paper warrants acceptance, provided the authors address the reviewers' concerns and further clarify the details of their analysis.